# Non-Wilson-Fisher kinks of $O(N)$ numerical bootstrap: From the deconfined phase transition to a putative new family of CFTs

Yin-Chen He[1]⋆, Junchen Rong[2] and Ning Su[3]

**1** Perimeter Institute for Theoretical Physics, Waterloo, Ontario N2L 2Y5, Canada
**2** DESY Hamburg, Theory Group, Notkestraße 85, D-22607 Hamburg, Germany
**3** Institute of Physics, École Polytechnique Fédérale de Lausanne,
CH-1015 Lausanne, Switzerland

⋆ yinchenhe@perimeterinstitute.ca

## Abstract

It is well established that the $O(N)$ Wilson-Fisher (WF) CFT sits at a kink of the numerical bounds from bootstrapping four point function of $O(N)$ vector. Moving away from the WF kinks, there indeed exists another family of kinks (dubbed non-WF kinks) on the curve of $O(N)$ numerical bounds. Different from the $O(N)$ WF kinks that exist for arbitary $N$ in $2 < d < 4$ dimensions, the non-WF kinks exist in arbitary dimensions but only for a large enough $N > N_c(d)$ in a given dimension $d$. In this paper we have achieved a thorough understanding for few special cases of these non-WF kinks, which already hints interesting physics. The first case is the $O(4)$ bootstrap in 2d, where the non-WF kink turns out to be the $SU(2)_1$ Wess-Zumino-Witten (WZW) model, and all the $SU(2)_{k>2}$ WZW models saturate the numerical bound on the left side of the kink. This is a mirror version of the $Z_2$ bootstrap, where the 2d Ising CFT sits at a kink while all the other minimal models saturating the bound on the right. We further carry out dimensional continuation of the 2d $SU(2)_1$ kink towards the 3d $SO(5)$ deconfined phase transition. We find the kink disappears at around $d = 2.7$ dimensions indicating the $SO(5)$ deconfined phase transition is weakly first order. The second interesting observation is, the $O(2)$ bootstrap bound does not show any kink in 2d ($N_c = 2$), but is surprisingly saturated by the 2d free boson CFT (also called Luttinger liquid) all the way on the numerical curve. The last case is the $N = \infty$ limit, where the non-WF kink sits at $(\Delta_\phi, \Delta_T) = (d-1, 2d)$ in $d$ dimensions. We manage to write down its analytical four point function in arbitrary dimensions, which equals to the subtraction of correlation functions of a free fermion theory and generalized free theory. An important feature of this solution is the existence of a full tower of conserved higher spin current. We speculate that a new family of CFTs will emerge at non-WF kinks for finite $N$, in a similar fashion as $O(N)$ WF CFTs originating from free boson at $N = \infty$.

# 1 Introduction

Conformal field theory (CFT) is of fundamental importance and has applications in various fields of physics, ranging from AdS/CFT in string theory to phase transitions in condensed matter physics. Bootstrap [1, 2], a technique utilizing intrinsic consistencies and constraints from the conformal symmetry, is one of most powerful tools in the study of conformal field theories. In two dimensions, thanks to the special Virasoro symmetry and Kac-Moody symmetry, bootstrap provides exact solutions of many CFTs including the 2d Ising CFTs and minimal model in 1980s [3]. However, for decades there was little progress of applying bootstrap to higher dimensional ($d > 2$) CFTs until the seminal work [4], which initiated the modern revival of the bootstrap method aiming at solving known CFTs (e.g. Wilson-Fisher (WF), QED, QCD, etc.) in higher dimensions, as well as exploring the uncharted territory of CFTs. In certain examples, the bootstrap method was used to extract the world's most precise predictions of critical exponents [5–12] of known CFTs. Many other successful applications were summarised in a recent review [13]. It is also possible that the bootstrap method can help us make progress on another frontier, namely discovering new CFTs.

Interesting CFTs usually sit at "kinks" of the bootstrap curve, such as the Ising model [14], the three dimensional $O(N)$ vector models [15] and many Wilson-Fisher CFTs with flavor symmetry groups to be subgroups of $O(N)$ [16–20]. Sometimes bootstrap curves shows more than one kink [19–23] [1]. For example, on the $O(N)$ bootstrap curve there are at least two kinks, the first one was successfully identified as $O(N)$ WF CFTs, while the nature of the second kink (we dub non-WF kink) remains an open question [3]. For a given space-time dimensions $d$, typically the non-WF kinks only appear when $N$ is larger than a critical $N_c$ [22]. In this paper, we focus on the study of the physics of non-WF kinks, and in some special cases we have achieved a thorough understanding analytically and numerically. These include the $O(4)$

---

[1]The non WF kinks of the O(N) bootstrap curves were first discovered in 2d in [21]. Similar results were discovered in higher dimensions in [2].

[3]See [19, 22] for attempts in identifying these non Wilson-Fisher kinks.

bootstrap kink in two space time dimensions, and the $N \to \infty$ limit in arbitrary dimensions. Even though the $O(2)$ bootstrap curve in two dimensions does not develop a kink, we find that the numerical bound is saturated by the free boson theory, which is also called Luttinger liquid in condensed matter literatures.

The 2d $O(4)$ non-WF kink turns out to be the $SU(2)_1$ Wess-Zumino-Witten (WZW) theory [24], and we find its dimensional continuation shows an interesting connection to the deconfined quantum critical point (DQCP) [25, 26]. The DQCP was originally proposed to describe a phase transition between two different symmetry breaking phases, namely Neel magnetic ordered state and valence bond state. Its critical theory has many dual descriptions [27], one of which is 3d $SO(5)$ non-linear sigma model (NL$\sigma$M) with level-1 WZW term. There is a long debate on whether DQCP is continuous or weakly first order [28–33]. Monte Carlo simulations are consistent with a continuous phase transition, but also show abnormal finite size scaling behaviors [32, 33]. More importantly, the critical exponent $\eta$ from Monte Carlo violates the rigorous bound from conformal bootstrap [13, 34], which dashes the hope of a continuous phase transition if $SO(5)$ symmetry is emergent. An interesting proposal to reconcile these inconsistencies is, DQCP is slightly complex (non-unitary) [27, 35, 36], hence shows pseudo-critical (weakly first order) behaviors. More concretely, a way to study the pseudo-critical behaviors is through dimensional continuation from 2d to 3d [37, 38]. The scheme of this dimensional continuation is motivated by the connection between DQCP and $SU(2)_1$ WZW theory: the former can be described by a 3-dimensional $SO(5)$ NL$\sigma$M with a level-1 WZW therm, while the latter is a 2-dimensional $SO(4)$ NL$\sigma$M with a level-1 WZW term. The action in integer dimensions can be written as

$$S = \int dx^d \frac{1}{2g^2} (\partial_\mu \vec{n}) \cdot (\partial^\mu \vec{n}) + k\Gamma^{WZW}[\vec{n}]]. \tag{1}$$

The scalar field $\vec{n}$ has $d + 2$ conponents, and satisfies the constraint $\vec{n} \cdot \vec{n} = 1$. Here $\Gamma^{WZW}$ is the standard Wess-Zumino-Witten term. Notice $\pi_{2+1}(S^3) = \pi_{3+1}(S^4) = \mathbf{Z}$, the level $k$ takes integer values. Naively, a physically plausible (though may not be mathematically concrete) way of dimensional continuation is to consider $d = 2 + \epsilon$ dimensional $SO(4 + \epsilon)$ NL$\sigma$M with a level-1 WZW therm. This maybe seems impossible in the action level, it is however not hard to study this scheme using numerical bootstrap. We study $O(4 + \epsilon)$ bootstrap in $d = 2 + \epsilon$, and observe that the kinks disappear at around $d^* = 2.7$. This agrees reasonably with the one-loop value $d^* = 2.77$ [37] and supports the scenario that the $SO(5)$ DQCP is weakly first order (pseudo-critical).

The solution of the $O(N = \infty)$ non-WF kink is more exotic. It turns out to be equal to the superposition of two physical four point function, for example, in $d = 3$ dimensions,

$$\frac{1}{2} \langle \bar{\psi}_i \eta(x_1) \bar{\psi}_j \eta(x_2) \bar{\psi}_k \eta(x_3) \bar{\psi}_l \eta(x_4) \rangle - \langle \phi_i(x_1) \phi_j(x_2) \phi_k(x_3) \phi_l(x_4) \rangle_{GFF}, \tag{2}$$

where $\psi_i$ are $N$ free Majorana fermions carrying $O(N)$ vector index, $\eta$ is another Majorana fermion that is neutral under $O(N)$ transformation, and $\phi_i$ is a scalar operator with scaling dimension $\Delta_\phi = 2$[4]. The bracket $\langle \ldots \rangle_{GFF}$ means the four point function of generalised free field (GFF) theory, or in other words, the four point function is calculated using Wick contraction. The exotic structure of subtracting two four point functions at $N = \infty$ limit makes it difficult to interpret finite-$N$ non-WF kinks as known CFTs. An important property of the solution at $N = \infty$ limit is, there exists a full tower of conserved higher spin current, a feature reminiscent of the free fermion theory. Therefore, it is possible that the non-WF kinks at finite

---

[4]We thank Zhijin Li and Andreas Stergio to suggest the possibility that this kink could be related to the free fermion theory.

$N$ become a new family of CFTs in a similar manner of $O(N)$ WF CFTs originating from the free boson theory.

The paper is organised in the following way. In Sec. 2 we discuss the general features of the non-WF kinks. In Sec. 3, we discuss the dimension continuation of the 2d $O(4)$ non-WF kink which corresponds the $SU(2)_1$ WZW model and its dimensional continuation. In the subsequent section, we discuss the $O(2)$ bootstrap bounds in two dimensions and the infinite-$N$ limit of $O(N)$ bootstrap. The plots in the paper are all calculated with $\Lambda = 27$ (the number of derivatives included in the numerics). For the definition of $\Lambda$ and other bootstrap parameters, we refer to [39].

*Note added.* After the completion of this work, we became aware of a parallel paper [40] which has some overlap with ours.

## 2 Non-WF kinks on the $O(N)$ bootstrap curve

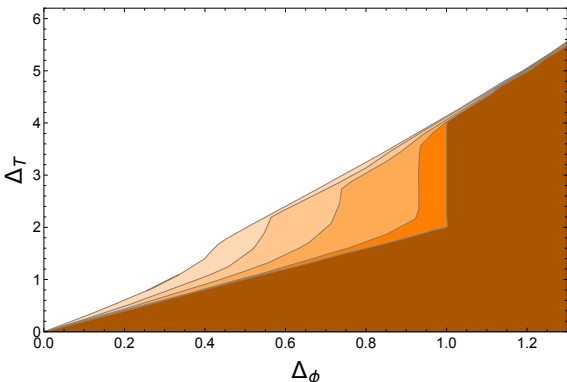

Figure 1: Bounds on $\Delta_T$ (the scaling dimension of the leading scalar operator in the rank-2 symmetric traceless tensor representation of $O(N)$) in terms of $\Delta_\phi$ of 2d CFTs with $O(3)$, $O(5)$, $O(10)$, $O(48)$, and $O(\infty)$ global symmetries (from left to right). Shaded regions are consistent with bootstrap constraints, therefore allow unitary CFTs to exist.

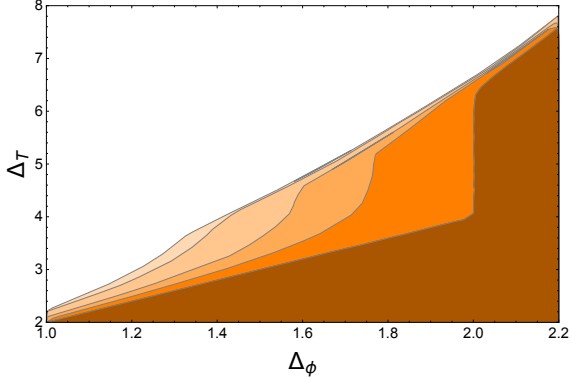

Figure 2: Bounds on $\Delta_T$ (the scaling dimension of the leading scalar operator in the rank-2 symmetric traceless tensor representation of $O(N)$) in terms of $\Delta_\phi$ of 3d CFTs with $O(16)$, $O(20)$, $O(40)$, $O(100)$, and $O(\infty)$ global symmetries (from left to right). Shaded regions are consistent with bootstrap constraints, therefore allow unitary CFTs to exist.

We start by considering the 4-point correlation function $\langle \phi_i(x_1)\phi_j(x_2)\phi_k(x_3)\phi_l(x_4)\rangle$ of a CFT with $O(N)$ global symmetry, with operator $\phi_i(x_1)$ carrying $O(N)$ vector index, and calculating it using the $\phi_a(x_1) \times \phi_b(x_2)$ OPE:

$$\phi_a \times \phi_b = S^+ + T^+ + A^- . \tag{3}$$

Here $S$, $T$ and $A$ refer to the operators in the $O(N)$ singlet, symmetric rank-2 tensor, and anti-symmetric rank-2 tensor. The superscript "$\pm$" denotes the spin selection: the $S$ and $T$ sectors contain even spin operators, while the $A$ sector contains only odd spin operators. The 4-point function from the s-channel decomposition is [41, 42],

$$
\begin{aligned}
\langle \phi_i(x_1)\phi_j(x_2)\phi_k(x_3)\phi_l(x_4)\rangle &= \frac{1}{x_{12}^{2\Delta_\phi}x_{34}^{2\Delta_\phi}}\Bigg[ \delta_{ij}\delta_{kl} \sum_{O\in S^+} \lambda_{\phi\phi O}^2 g_{\Delta,l}(u,v) \\
&\quad + (\frac{1}{2}\delta_{il}\delta_{jk} + \frac{1}{2}\delta_{ik}\delta_{jl} - \frac{1}{N}\delta_{ij}\delta_{kl}) \sum_{O\in T^+} \lambda_{\phi\phi O}^2 g_{\Delta,l}(u,v) \\
&\quad + (\frac{1}{2}\delta_{il}\delta_{jk} - \frac{1}{2}\delta_{ik}\delta_{jl}) \sum_{O\in A^-} \lambda_{\phi\phi O}^2 g_{\Delta,l}(u,v) \Bigg] .
\end{aligned}
\tag{4}
$$

$g_{\Delta,l}(u,v)$ is the conformal block, and $u = x_{12}^2 x_{34}^2/(x_{24}^2 x_{13}^2)$, $v = x_{14}^2 x_{23}^2/(x_{24}^2 x_{13}^2)$. Similarly, by considering the four point in the crossed channel one can get another conformal block decomposition of the 4-point correlation function, which is Eq. (4) with $i \leftrightarrow k$ and $x_1 \leftrightarrow x_3$. Equating two different channels one obtains a non-trivial crossing symmetric equation [41, 42].

$$
\sum_{O\in S^+} \lambda_{\phi\phi O}^2 \begin{pmatrix} F \\ -H \\ 0 \end{pmatrix} + \sum_{O\in T^+} \lambda_{\phi\phi O}^2 \begin{pmatrix} \frac{F(N-2)}{2N} \\ \frac{H(N+2)}{2N} \\ \frac{F}{2} \end{pmatrix} + \sum_{O\in A^-} \lambda_{\phi\phi O}^2 \begin{pmatrix} \frac{F}{2} \\ \frac{H}{2} \\ -\frac{F}{2} \end{pmatrix} = 0 , \tag{5}
$$

with

$$F = v^{\Delta_\phi} g_{\Delta,l}(u,v) - u^{\Delta_\phi} g_{\Delta,l}(v,u), \quad H = v^{\Delta_\phi} g_{\Delta,l}(u,v) + u^{\Delta_\phi} g_{\Delta,l}(v,u).$$

By demanding the OPE coefficients $\lambda_{\phi\phi O}$ to be real, from the bootstrap equation one can obtain numerical bounds of scaling dimensions of operators in the $\phi \times \phi$ OPE, in terms of $\phi$'s scaling dimension $\Delta_\phi$ [4]. Typically one will bound the lowest scaling dimensions (e.g. $\Delta_S$, $\Delta_T$) of scalar operators in different channels of group representations. It is well known that the $O(N)$ WF CFT appears at kinks on the curve of numerical bounds of $\Delta_S$ and $\Delta_T$ in $d = 3$ dimensions [15]. The result can be easily generalized to $2 < d < 4$. Besides the $O(N)$ WF there are also other kinks (i.e. non-WF kinks), which, for example, are shown in Fig. 1 and Fig. 2. These non-WF kinks exist on both the $\Delta_\phi - \Delta_S$ and $\Delta_\phi - \Delta_T$ curve, and it seems that the kinks on two curves have identical $\Delta_\phi$. We find that the bounds of $\Delta_T$ converge faster than those of $\Delta_S$. Also as will be clear later, in most cases it is more physically meaningful to study the $\Delta_\phi - \Delta_T$ curve rather than the $\Delta_\phi - \Delta_S$ curve.

Different from the $O(N)$ WF kinks which only occur in $2 < d < 4$ dimensions, the non-WF kinks seem to exist in arbitrary dimensions ($2 \le d \le 6$ at least). Also in $2 < d < 4$ dimensions, the positions of non-WF kinks are quite far away from WF kinks. For example, in $d = 3$ dimension (see Fig. 2) non-WF kinks have $(\Delta_\phi, \Delta_T) \approx (1.2 \sim 2.0, 3.8 \sim 6.0)$ (as $N$ varies), while WF kinks are pretty close to the Gaussian theory with $(\Delta_\phi, \Delta_T) \approx (0.5 \sim 0.52, 1.0 \sim 1.3)$. Another crucial feature is, for a given space-time dimension $d$ the non-WF kinks only appear when $N$ is larger than a critical $N_c$ [22]. In $d = 2$ dimension $N_c = 2$, and $N_c$ seems to increase with $d$ [5]. Also the kink becomes sharper as $N$ increases (see Fig. 1 and Fig. 2), and in the $N \to \infty$ limit the kink evolves into a sudden jump at $(\Delta_\phi, \Delta_T) = (d-1, 2d)$.

---

[5]It is also worth mentioning that the numerical convergence is slower for a larger $d$.

In general, except for a few cases, it is unclear if these non-WF kinks as well as the numerical bounds have any relation to CFTs or any physical theories. The rest of the paper will discuss several special cases where we have good understanding, through which we hope to inspire the understanding of non-WF kinks in general cases.

## 3 From 2d $SU(2)_1$ WZW to 3d $SO(5)$ DQCP

### 3.1 $O(4)$ symmetry in 2d: $SU(2)_k$ WZW theory

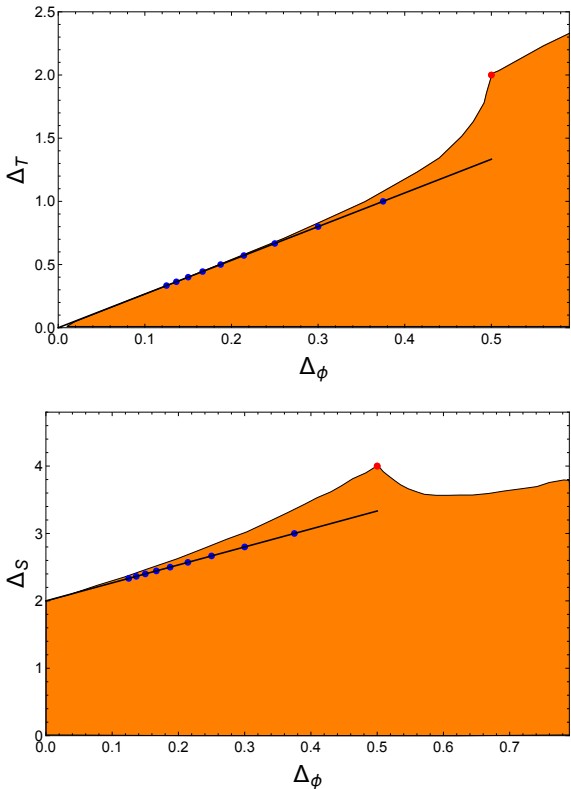

Figure 3: Numerical bounds of $\Delta_S$ (the scaling dimension of the leading scalar operator in singlet representation of $O(4)$) and $\Delta_T$ (the scaling dimension of the leading scalar operator in the rank-2 symmetric traceless tensor representation of $O(4)$) of $O(4)$ CFTs in 2d. Shaded regions are consistent with bootstrap constraints, therefore allow unitary CFTs to exist. The scaling dimensions of $SU(2)_k$ WZW theory are $(\Delta_\phi, \Delta_T, \Delta_S) = (\frac{3}{2(k+2)}, \frac{8}{2(k+2)}, 2 + \frac{8}{2(k+2)})$ for $k \geq 2$, which is denoted as blue dots connected by a solid line. The $k = 1$ theory, located at $(\Delta_\phi, \Delta_T, \Delta_S) = (0.5, 2, 4)$, is denoted as the red dot.

The $SU(2)_k$ WZW theory has a $SO(4) \cong \frac{SU(2)_L \times SU(2)_R}{Z_2}$ global symmetry, and a special parity which flips one space direction and the two $SU(2)$ groups simultaneously. It turns out that a subset of the crossing equation which equals (5) at $N = 4$ is already sufficient for detecting $SU(2)_k$ WZW models (see Appendix A for more discussion on this). Fig. 3 shows the numerical bound for the leading singlet ($S$) and rank-2 tensor ($T$), which has a kink at $(\Delta_\phi, \Delta_S) = (0.5, 4)$ and $(\Delta_\phi, \Delta_T) = (0.5, 2)$, respectively. They match the theoretical values of $SU(2)_{k=1}$ WZW theory. More interestingly, $SU(2)_{k\geq 2}$ WZW theory seems to saturate the numerical bound of $\Delta_T$ on the left side of $SU(2)_{k=1}$ WZW theory. This phenomena is a mirror version of well-

known observation of the $Z_2$ bootstrap in 2d, in which the 2d Ising CFT appears at the kink and all the minimal models saturate the numerical bound on the right hand side of Ising CFT [4,43]. The reason that $SU(2)_1$ WZW appears as a kink is the leading operator in the $T$-channel of $SU(2)_k$ WZW gets decoupled from the theory at $k = 1$. On the other hand, the numerical bound of $\Delta_S$ seems to be larger than the $SU(2)_{k \geq 2}$ WZW theory. It is unclear whether it is a convergence issue, although we do not see a visible improvement from $\Lambda = 19$ to $\Lambda = 27$.

One can further read out the spectrum of $S, T$ and $A$ channel operators from the extremal functional method [44]. It is also possible to numerically study the OPE's of the leading operators of each channel. We found that the spectrum and OPE coefficients of the solution at the kink agrees with the $SU(2)_1$ WZW theory.

## 3.2   Dimensional continuation to $SO(5)$ DQCP

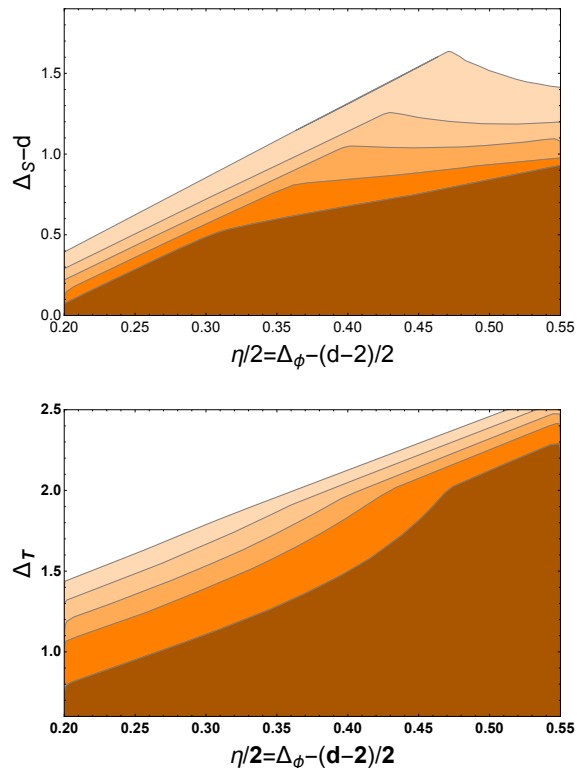

Figure 4: The numerical bounds for $\Delta_T$ (the scaling dimension of the leading scalar operator in the rank-2 symmetric traceless tensor representation) and $\Delta_S$ (the scaling dimension of the leading scalar operator in singlet representation) for CFTs with $O(4 + \epsilon)$ symmetry in $d = (2 + \epsilon)$ dimensions. Shaded regions are consistent with bootstrap constraints, therefore allow unitary CFTs to exist. The plots correspond to $\epsilon = 0.2, 0.4, 0.5, 0.6, 0.7$ from top (below) to below (top) for $\Delta_S$ ($\Delta_T$).

The dimensional continuation of the WF kinks has been explored before [45], and it was found the scaling dimensions at the WF kinks in fractional dimensions $2 < d < 4$ are in agreement with the $\epsilon$-expansion calculation. In this section we will study an exotic way of dimensional continuing the non-WF kink, motivated by recent papers [37,38] that studied the deconfined quantum critical point (DQCP) [25,26].

As shown in previous section, the $SU(2)_1$ WZW theory appears as a kink in the curve of $O(4)$ bootstrap bounds in $d = 2$ dimensions, so we can further bootstrap $O(4 + \epsilon)$ symmetry in $d = 2 + \epsilon$ dimensions. As shown in Fig. 4, for small $\epsilon$ the kink still exists, but becomes

weaker and weaker as $\epsilon$ increases, and finally disappears around $\epsilon^* \approx 0.7$ [6] This reasonably agrees with the one-loop value $\epsilon^* = 0.77$ [37]. $\eta = 2\Delta_\phi - d + 2$ and $\Delta_S - d$ decreases with $\epsilon$, which is also consistent with the expectation of pseudo-critical behavior. Theoretically, the CFT can become complex when the lowest singlet operator becomes relevant [27, 35, 36]. In our numerical data, however, $\Delta_S$ seems to be larger than $d$ when $\epsilon = 0.7$. This might be an artifact of numerical convergence, also it is hard to locate the precise critical $\epsilon^*$ as the kink becomes very weak.

# 4 Analytical results for some other bootstrap bounds

## 4.1 $O(2)$ symmetry: 2d free boson/ Luttinger liquid

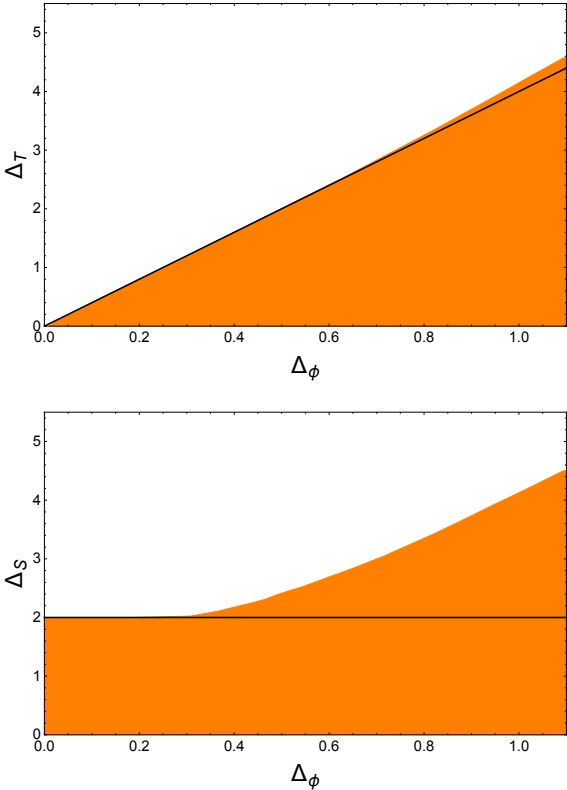

Figure 5: The numerical bounds for $\Delta_T$ (the scaling dimension of the leading scalar operator in the rank-2 symmetric traceless tensor representation of $O(2)$) and $\Delta_S$ (the scaling dimension of the leading scalar operator in singlet representation of $O(2)$) for CFTs with $O(2)$ symmetry in 2d. Shaded regions are consistent with bootstrap constraints, therefore allow unitary CFTs to exist. The solid line corresponds to 2d free boson which has $\Delta_T = 4\Delta_\phi$ and $\Delta_S = 2$.

The numerical bounds from $O(2)$ bootstrap does not show any kink [7], but it indeed detects 2d CFTs, namely a 2d free boson (also called Luttinger liquid in condensed matter literatures).

---

[6]The critical $\epsilon^*$ is read out from the $\Delta_\phi - \Delta_S$ curve as the kink is sharper there. Notice a subtlety when we apply bootstrap method to study conformal field theories in factional dimensions is that these theories are intrinsically non-unitary, due to negative norm states [46, 47]. Such non-unitary states have high scaling dimensions and the bootstrap results are insensitive to them. The disappearance of the kink at $\epsilon^* \approx 0.7$, on the other hand, should be explained by the fixed point annihilation mechanism proposed in [37, 38].

[7]In 2d the non-WF kink appears only when $N > 2$.

It is well known that the 2d free boson is a CFT with an exact marginal operator. Its global symmetry is $U(1)_L \times U(1)_R$, but we can just consider its diagonal $U(1)$, i.e. the charge conservation symmetry. The charge creation operator (i.e. vertex operator), $e^{i\alpha\Phi}$, can be written as a $O(2)$ vector $(\phi_1, \phi_2) = (\text{Re}(e^{i\alpha\Phi}), \text{Im}(e^{i\alpha\Phi}))$. Its scaling dimension $\Delta_\phi$ can be continuously tuned from 0 to $\infty$ by deforming the compactification radius of bosons. The lowest scaling dimension in the $T$-channel is $\Delta_T = 4\Delta_\phi$, while in the $S$-channel one has $\Delta_S = 2$ independent of $\Delta_\phi$. The four point function is,

$$
\langle \phi_i(x_1)\phi_j(x_2)\phi_k(x_3)\phi_l(x_4) \rangle = \frac{1}{x_{12}^{2\Delta_\phi} x_{34}^{2\Delta_\phi}} \Big[ \delta_{ij}\delta_{kl}(v^{-\Delta_\phi} + v^{\Delta_\phi})
$$
$$
+ (\delta_{il}\delta_{jk} + \delta_{ik}\delta_{jl} - \delta_{ij}\delta_{kl})u^{2\Delta_\phi}v^{-\Delta_\phi}
$$
$$
+ (\delta_{il}\delta_{jk} - \delta_{ik}\delta_{jl})(v^{-\Delta_\phi} - v^{\Delta_\phi}) \Big] . \tag{6}
$$

Fig. 5 shows the numerical bounds of $\Delta_T$ and $\Delta_S$ in terms of $\Delta_\phi$. In the $\Delta_\phi - \Delta_T$ curve it is clear that the 2d free boson saturates the numerical bounds. For large $\Delta_\phi$ there is a small discrepancy due to the numerical error of finite $\Lambda$. The $\Delta_\phi - \Delta_S$ curve, on the other hand, is only saturated by the 2d free boson at small $\Delta_\phi$. At large $\Delta_\phi$ the numerical bounds approach the point $(\Delta_\phi, \Delta_S) = (1, 4)$, which corresponds to the four point function (7) to be discussed later. This result again suggests that the $\Delta_\phi - \Delta_T$ curve is more intrinsic for understanding the non-WF physics in the $O(N)$ bootstrap calculation.

## 4.2 Infinite-$N$ limit

The infinite-$N$ limit can be studied directly by taking $1/N = 0$ in the bootstrap equation. In $d$ dimensions the kink sits at $(\Delta_\phi, \Delta_T) = (d-1, 2d)$, and on the left of the kink the GFF saturates numerical bounds $\Delta_T = 2\Delta_\phi$. The $S$-sector spectrum of $\phi \times \phi$ OPE is very exotic: the scalar channel ($l = 0$) is totally empty with no operator present (except the identity operator), while in other spin ($l > 0$) channel only one operator, i.e. the higher spin conserved current ($\Delta_{S,l} = l + d - 2$), is present for each $l$. The 4-point correlation function at the kink turns out to be [8],

$$
\langle \phi_i(x_1)\phi_j(x_2)\phi_k(x_3)\phi_l(x_4) \rangle = \frac{1}{|x_{12}x_{34}|^{-2(d-1)}}
$$
$$
\times (\delta_{ij}\delta_{kl}G^a[u,v] + \delta_{il}\delta_{jk}G^b[u,v] + \delta_{ik}\delta_{jl}G^c[u,v]), \tag{7}
$$
$$
G^a[u,v] = 1 - \frac{u^{d/2-1}(-1 + u + v + v^{d/2} + uv^{d/2} - v^{d/2+1})}{2v^{d/2}},
$$
$$
G^b[u,v] = u^{d-1}v^{1-d}G^a[v,u],
$$
$$
G^c[u,v] = G^b[u/v, 1/v].
$$

This four point function is unitary [9]. Surprisingly, The above four point function equals to the subtraction of correlation functions of two different theories, namely a free fermion theory (FFT) and a GFF theory: in $d = 3$ dimensions, where (7) equals

$$
\frac{1}{2}\langle \bar{\psi}_i\eta(x_1)\bar{\psi}_j\eta(x_2)\bar{\psi}_k\eta(x_3)\bar{\psi}_l\eta(x_4) \rangle - \langle \phi_i(x_1)\phi_j(x_2)\phi_k(x_3)\phi_l(x_4) \rangle_{GFF}. \tag{8}
$$

The FFT contains $N$ free Majorana fermions $\psi_i$ and a single free Majorana fermion $\eta$, so the fermion bilinear $\bar{\psi}_i\eta$ is a $O(N)$ vector. Using Wick contraction, the above expression reduces

---

[8]Notice this four point function can also be viewed as a solution to the O(N) bootstrap equations even at finite N. The point $(\Delta_\phi, \Delta_T) = (d-1, 2d)$ almost saturates the finite N bootstrap bound.

[9]In two and four dimensions, by expanding the four point function in conformal blocks, we have proven that for each channel, the operators have positive OPE².

to products of two point functions

$$\langle \bar{\psi}_i(x_1)\psi_j(x_2) \rangle = \delta_{ij} \frac{x_{12}^{\mu} \gamma_{\mu}}{|x_{12}|^3}, \quad \langle \bar{\eta}(x_1)\eta(x_2) \rangle = \frac{x_{12}^{\mu} \gamma_{\mu}}{|x_{12}|^3}, \quad \text{and} \quad \langle \phi(x_1)\phi(x_2) \rangle = \frac{1}{|x_{12}|^4} . \quad (9)$$

With a few lines of algebra one can show that (7) and (8) are identical.

The solution (7) has quite a few exotic features. First of all, if we bound the $\lambda^2_{\phi\phi T_{\mu\nu}}$ OPE coefficient numerically, we will get that the central charge $c = c_f$. Here $c_f$ is the central charge of a single Majorana fermion. Its spectrum also contains conserved higher spin currents. This poses a puzzle that the theory seemingly contradicts a theorem [48] saying that CFTs with conserved higher spin currents are free theories which have a central charge proportional to $N$. From (8), the solution to this puzzle is clear. The theory contains more than one conserved spin-2 current,

$$T^1_{\mu\nu} = \bar{\psi}_i \gamma_{[\mu} \partial_{\nu]} \psi_i, \quad \text{and} \quad T^2_{\mu\nu} = \bar{\eta} \gamma_{[\mu} \partial_{\nu]} \eta, \quad (10)$$

while the theorem in [48] assumed a single spin-2 current. Notice

$$\lambda^2_{\phi\phi T^1_{\mu\nu}} \sim \frac{1}{N}, \quad \text{and} \quad \lambda^2_{\phi\phi T^2_{\mu\nu}} \sim \frac{1}{c_f} . \quad (11)$$

Only the contribution of $\lambda^2_{\phi\phi T^2}$ survives in the large $N$ limit. We can also think about what kind of $\frac{1}{N}$ corrections that will turn (7) into a "good" CFT. By "good" we mean CFTs with a single conserved current and order $N$ central charge. This is possible if $T^2_{\mu\nu}$ acquires anomalous dimension. The second exotic feature is the minus sign in front of the GFF four point function in (7), this makes the interpretation of it as known CFTs really difficult. Another exotic feature is that if we decompose the four point function (7) into conformal blocks, we will find that there is no spin-0 block in the S-channel. This is also observed numerically. It turns out that the OPE coefficients of $S$-channel scalars of both FFT and GFF scales as $\mathcal{O}(1/N)$, therefore disappears at the strict $N = \infty$ limit. We also observe that the spectrum of GFF is a subset of the spectrum of FFT. Consequently, a four point correlation function $c_1 \langle 4pt \rangle_{FFT} - c_2 \langle 4pt \rangle_{GFF}$ is consistent with bootstrap as long as the OPE coefficients $c_1 \lambda^2_{FFT} - c_2 \lambda^2_{GFF}$ are positive for all the operators in GFF. More importantly, by choosing $c_1, c_2$ properly ($c_1 = 1/2$, $c_2 = 1$), many operators disappear in the block expansion. These includes the $(\Delta, l) = (2d-2, 0)$ operator in the $T$-channel and many other operators. After this superposition, the leading scalar operator in the $T$-channel has scaling dimension to be $(\Delta, l) = (2d, 0)$. This explains why the numerical bound follows $\Delta_T = 2\Delta_\phi$ (GFF) for small $\Delta_\phi$, and has a sudden jump at $\Delta_\phi = d - 1$ from $\Delta_T = 2d - 2$ to $\Delta_T = 2d$. Since FFT and GFF are present in arbitrary dimension, we expect the non-WF kink in the infinite-$N$ limit to also exist in arbitrary dimensions, and we have numerically verified it for $2 \leq d \leq 6$ dimensions.

This teaches us an important lesson, a kink on the bootstrap curve can correspond to the subtraction of four point functions of two different theories. The key requirement for this to happen is the spectrum of one theory is a subset of the spectrum of the other theory. This requirement is apparently very stringent in $d > 2$ dimensions, namely except for (generalized) free theories there is no known pair of theories satisfying it. On the other hand, the non-WF kinks at finite $N$ obviously do not correspond to free theories. Therefore, it would be interesting and exotic if the appearance of non-WF kinks at finite $N$ are also due to the subtraction of four point functions of two theories.

The other possibility is that non-WF kinks detect a single theory rather than the subtraction of two theories. The four point function in the infinite-$N$ limit, on the other hand, just happens to be identical to the subtraction of FFT and GFF. Previous identification of 2d $SU(2)_1$ WZW theory as the $O(4)$ non-WF kink seems to favor this scenario.

# 5 Conclusion

We study the non-WF kinks in the $O(N)$ bootstrap curves. This family of kinks, different from the WF kink, exists in arbitrary dimension. In a given dimension, there exists a critical $N_c$ below which the kink disappears. In general, we do not understand the physics of this new family of kinks except for few cases. In the infinite-$N$ limit, the kink sits at $(\Delta_\phi, \Delta_T) = (d-1, 2d)$ with $d$ being the space-time dimensions. The four point function at the kink equals to the subtraction of correlation functions of a free fermion theory and generalized free theory. One lesson from this example is, subtracting two theories (whose spectrum are similar) could also generate a kink in the curve of bootstrap bounds. However, it seems that the kink at finite $N$ cannot be interpreted in this way. For example, the $O(4)$ kink in 2d corresponds to the $SU(2)_1$ WZW theory. We further study the dimensional continuation of the $SU(2)_1$ WZW kink to 3d and discuss its relation with deconfined phase transitions.

Besides the kink, the numerical bounds in 2d also have a few intriguing properties. The $O(2)$ curve does not have a kink, but is saturated by the free boson theory ($\Delta_T = 4\Delta_\phi$), a CFT with continuously tunable scaling dimensions due to an exact marginal operator. On the $O(4)$ curve, the $SU(2)_1$ WZW theory appears at the kink and $SU(2)_{k>1}$ WZW theories ($\Delta_T = \frac{8}{3}\Delta_\phi$) saturate the numerical bounds on the left side of the kink. For a general $N$, the numerical bounds on the left side of the kink seems to obey a simple algebraic relation $\Delta_T = \frac{2N}{N-1}\Delta_\phi$. It will be interesting to know if there exists an analytical four point function giving this relation for a general $N$.

Except for few cases it is rather unclear which physical theories the non-WF kinks correspond to. A major challenge is that there is no known CFT whose symmetry and operator contents are similar to what we observed numerically at the non-WF kinks [10]. There was one proposal that the intrinsic symmetry of 3d non-WF kinks is $SU(N^*)$ rather than $O(N)$ (with $N \sim (N^*)^2 - 1$, and one should bootstrap the four point function of $SU(N^*)$ adjoint operators instead of $O(N)$ vector operators [22] [11]. In the 3d $SU(N^*)$ adjoint bootstrap, there appear two adjacent kinks on the bound of leading $SU(N^*)$ singlet operator, and they were interpreted as QED$_3$-Gross-Neveu and QED$_3$ CFTs respectively, while the $SU(N^*)$ adjoint scalar field $\phi$ is interpreted as the fermion bilinear operator. [12] This proposal is interesting however one should be particularly careful about the following: Firstly, the scaling dimension of $SU(N^*)$ singlet at the kink is way larger than that of QED$_3$ (e.g. the kink of $SU(15)$ has $\Delta_S \sim 10$ but the $N_f = 15$ QED$_3$ has $\Delta_S < 4$). A plausible but unsettling possibility is the numerical convergence is extremely slow due to that the OPE coefficient is small. Secondly, at large enough $N$, QED$_3$-Gross-Neveu has a relevant singlet (i.e. the mass term of Yukawa field $\phi^2$ with $\Delta_S = 2 + O(1/N^*)$) while the leading S-channel scalar operator of QED$_3$ is irrelevant (with $\Delta_S = 4 + O(1/N^*)$). Their the fermion bilinear operators have similar scaling dimensions, it is hard to imagine that they both saturate the bootstrap bound. It would be interesting to study the large $N$ limit so as to improve our understanding.

Although a thorough understanding of the non-WF kinks remains elusive, we think many of these kinks would have contact with physical theories given the presented results of $O(2)$, $O(4)$ at 2d and $O(\infty)$ at arbitrary dimensions. An exciting possibility is that they correspond

---

[10]Besides the $O(N)$ WF CFTs, QCD$_3$ with $O(N_c)$ gauge group also has $O(N)$ global symmetry. However, in such QCD$_3$ theories the low lying operators are fermion bilinears which are the $O(N)$ rank-2 tensors rather than the $O(N)$ vectors we considered here.

[11]The S-channel bootstrap bounds obtained by studying four-point functions of scalar operators in the $O(N)$ vector representations coincides with the S-channel bootstrap bounds obtained by studying four-point functions of scalar operators in the $SU(N^*)$ adjoint representation. See [40] for a proof of the coinciende of the bounds.

[12]Since the two kinks are so close to each other, it would be interesting to confirm that the two kinks indeed corresponds to two separate solutions of the crossing equation, possibly by studying the extremal functional with higher numerical precision.

to a new family of CFTs that were unknown before. To make progress it is necessary to obtain precise spectra of the putative CFTs, which might be achieved by studying the mixed correlator bootstrap of $O(N)$ vector $V$ and symmetric rank-2 tensor $T$.

## Acknowledgements

We are grateful to Chong Wang, Davide Gaiotto, Pedro Liendo, Volker Schomerus, Cenke Xu, Matthias Gaberdiel for stimulating discussions. J. R. and N. S. would like to thank the hospitality of Perimeter Institute while part of the work was finished. Research at Perimeter Institute is supported in part by the Government of Canada through the Department of Innovation, Science and Economic Development Canada and by the Province of Ontario through the Ministry of Colleges and Universities. N. S. is supported by the European Research Council Starting Grant under grant no. 758903 and Swiss National Science Foundation grant no PP00P2-163670. The work of J.R. is supported by the DFG through the Emmy Noether research group The Conformal Bootstrap Program project number 400570283. The numerics is solved using SDPB program [39] and simpleboot(https://gitlab.com/bootstrapcollaboration/simpleboot). This work used PI Symmetry cluster and EPFL SCITAS cluster.

## A  Bootstrapping the $SU(2)_k$ WZW theory: $O(4)$ versus $SO(4)$

The action (1) preserves a usual $SO(d+2)$ symmetry and a special kind of parity

$$\mathbb{P}^* = \{\mathbb{Z}_2^{O(d+2)} \times \mathbb{P}\}_{\text{diag}}. \tag{12}$$

One can choose its action on the scalar fields $\vec{n}$ to be

$$(n^1(x^\mu), n^2(x^\mu)\ldots, n^{d+2}(x^\mu)) \to (-n^1(\tilde{x}^\mu), n^2(\tilde{x}^\mu)\ldots, n^{d+3}(\tilde{x}^\mu)),$$
$$\tilde{x}^\mu = (x^0, -x^1, \ldots x^{d-1}). \tag{13}$$

Specialized to $SU(2)_1$ WZW models, the symmetry fixes the four point function to have the following form

$$
\begin{aligned}
\langle \phi_i(x_1)\phi_j(x_2)\phi_k(x_3)\phi_l(x_4)\rangle &= \frac{1}{|x_{12}|^{2\Delta_\phi}|x_{34}|^{2\Delta_\phi}} \\
&\times \Bigg( P_{ijkl}^{(0,0)} \sum_{O\in(0,0)} \lambda_{\phi\phi O}^2 \left(g_{\Delta,l}(z,\bar{z}) + g_{\Delta,-l}(z,\bar{z})\right) \\
&+ P_{ijkl}^{(1,1)} \sum_{O\in(1,1)} \lambda_{\phi\phi O}^2 \left(g_{\Delta,l}(z,\bar{z}) + g_{\Delta,-l}(z,\bar{z})\right) \\
&+ \sum_{O\in(0,1)+(1,0)} \lambda_{\phi\phi O}^2 \left(P_{ijkl}^{(1,0)} g_{\Delta,l}(z,\bar{z}) + P_{ijkl}^{(0,1)} g_{\Delta,-l}(z,\bar{z})\right) \Bigg).
\end{aligned}
\tag{14}
$$

The cross ratio is defined in two dimension as

$$z = \frac{z_{12}z_{34}}{z_{13}z_{24}}, \quad \bar{z} = \frac{\bar{z}_{12}\bar{z}_{34}}{\bar{z}_{13}\bar{z}_{24}}, \quad \text{with } z_{ij} = x_{ij}^0 + x_{ij}^1, \text{ and } \bar{z}_{ij} = x_{ij}^0 - x_{ij}^1. \tag{15}$$

The SO(4) projectors are defined as

$$
\begin{aligned}
P_{ijkl}^{(0,0)} &= \frac{1}{4}\delta_{ij}\delta_{kl}\,,\\
P_{ijkl}^{(1,1)} &= \frac{1}{2}\delta_{il}\delta_{jk} + \frac{1}{2}\delta_{ik}\delta_{jl} - \frac{1}{4}\delta_{ij}\delta_{kl}\,,\\
P_{ijkl}^{(0,1)} &= \frac{1}{4}\left(\delta_{il}\delta_{jk} - \delta_{ik}\delta_{jl}\right) + \frac{1}{4}\epsilon_{ijkl}\,,\\
P_{ijkl}^{(1,0)} &= \frac{1}{4}\left(\delta_{il}\delta_{jk} - \delta_{ik}\delta_{jl}\right) - \frac{1}{4}\epsilon_{ijkl}\,.
\end{aligned}
\tag{16}
$$

The conformal blocks are

$$
\begin{aligned}
g_{\Delta,l} &= k_{\Delta+l}(z)k_{\Delta-l}(\bar{z})\\
k_\beta(x) &= x^{\beta/2}{}_2F_1(\beta/2,\beta/2,\beta,x)\,.
\end{aligned}
\tag{17}
$$

Notice the parity symmetry (12) does the following interchanges

$$
P_{ijkl}^{(1,0)} \longleftrightarrow P_{ijkl}^{(0,1)},\quad z\longleftrightarrow\bar{z},\quad\text{and}\quad g_{\Delta,l}(z,\bar{z})\longleftrightarrow g_{\Delta,-l}(z,\bar{z})\,,
\tag{18}
$$

therefore fix the last term in (14). Let us rewrite (14) into the following form

$$
\langle\phi_i(x_1)\phi_j(x_2)\phi_k(x_3)\phi_l(x_4)\rangle = \frac{1}{|x_{12}||x_{34}|}\left(G_{ijkl}(z,\bar{z}) + G_{ijkl}^{(t)}(z,\bar{z})\right),
\tag{19}
$$

where

$$
\begin{aligned}
G_{ijkl}(z,\bar{z}) =\ & \frac{1}{4}\delta_{ij}\delta_{kl}\sum_{O\in(0,0)^+}\lambda_{\phi\phi O}^2\left(g_{\Delta,l}(z,\bar{z}) + g_{\Delta,-l}(z,\bar{z})\right)\\
& + \left(\frac{1}{2}\delta_{il}\delta_{jk} + \frac{1}{2}\delta_{ik}\delta_{jl} - \frac{1}{4}\delta_{ij}\delta_{kl}\right)\sum_{O\in(1,1)^+}\lambda_{\phi\phi O}^2\left(g_{\Delta,l}(z,\bar{z}) + g_{\Delta,-l}(z,\bar{z})\right)\\
& + \left(\frac{1}{2}\delta_{il}\delta_{jk} - \frac{1}{2}\delta_{ik}\delta_{jl}\right)\sum_{O\in(0,1)^-+(1,0)^-}\lambda_{\phi\phi O}^2\left(g_{\Delta,l}(z,\bar{z}) + g_{\Delta,-l}(z,\bar{z})\right),
\end{aligned}
\tag{20}
$$

and

$$
G_{ijkl}^{(t)}(z,\bar{z}) =\ \frac{1}{4}\epsilon_{ijkl}\sum_{O\in(0,1)+(1,0)}\lambda_{\phi\phi O}^2\left(g_{\Delta,l}(z,\bar{z}) - g_{\Delta,-l}(z,\bar{z})\right).
\tag{21}
$$

$G_{ijkl}(z,\bar{z})$ is invariant under the usual parity transformation, while $G_{ijkl}^{(t)}(z,\bar{z})$ is only invariant under the twisted parity (12). As is clear from the invariant tensor, they satisfy the crossing equation independently.

The parity even combination of the block

$$
g_{\Delta,l}(z,\bar{z}) + g_{\Delta,-l}(z,\bar{z}),
\tag{22}
$$

can be dimensional continued to $d > 2$, while the parity odd combination

$$
g_{\Delta,l}(z,\bar{z}) - g_{\Delta,-l}(z,\bar{z}),
\tag{23}
$$

can not. This can be shown by solving the Casimir equation directly. Another way to understand this is that in $d = 2$, the rotation group is $SO(2)$, the spin $l$ state and spin $-l$ state are

two independent irreducible representation of the conformal group. Their blocks $g_{\Delta,l}(z,\bar{z})$ and $g_{\Delta,-l}(z,\bar{z})$ (hence (22) and (23)) appear independently in the four point function. In higher dimensions, however, they belong to the same irreducible representation. There is a unique block. We can derive the crossing equation from (14),

$$\sum_{O\in(0,0)^+}\lambda_{\phi\phi O}^2\begin{pmatrix}\frac{1}{4}F\\-\frac{1}{4}H\\0\\0\end{pmatrix}+\sum_{O\in(1,1)^+}\lambda_{\phi\phi O}^2\begin{pmatrix}\frac{F}{4}\\\frac{3H}{4}\\\frac{F}{2}\\\frac{1}{4}\mathcal{H}\end{pmatrix}+\sum_{O\in(1,0)^-+(0,1)^-}\lambda_{\phi\phi O}^2\begin{pmatrix}\frac{F}{2}\\\frac{H}{2}\\-\frac{F}{2}\\0\end{pmatrix}=0\,,\quad(24)$$

with

$$\begin{aligned}F&=((1-z)(1-\bar{z}))^{\Delta_\phi}\\&\quad\times(g_{\Delta,l}(z,\bar{z})+g_{\Delta,-l}(z,\bar{z}))-(z\bar{z})^{\Delta_\phi}(g_{\Delta,l}(1-z,1-\bar{z})+g_{\Delta,-l}(1-z,1-\bar{z}))\\H&=((1-z)(1-\bar{z}))^{\Delta_\phi}\\&\quad\times(g_{\Delta,l}(z,\bar{z})+g_{\Delta,-l}(z,\bar{z}))+(z\bar{z})^{\Delta_\phi}(g_{\Delta,l}(1-z,1-\bar{z})+g_{\Delta,-l}(1-z,1-\bar{z}))\\\mathcal{H}&=((1-z)(1-\bar{z}))^{\Delta_\phi}\\&\quad\times(g_{\Delta,l}(z,\bar{z})-g_{\Delta,-l}(z,\bar{z}))+(z\bar{z})^{\Delta_\phi}(g_{\Delta,l}(1-z,1-\bar{z})-g_{\Delta,-l}(1-z,1-\bar{z}))\,.\end{aligned}\quad(25)$$

The last row of the crossing equation (24) comes from the twist parity invariant part $G_{ijkl}^{(t)}$. Since we do not know how to dimensional continue it to higher dimension, we will discard this line when doing numerical bootstrap. This truncation can also be viewed as originated form the fact that the invariant tensor $\epsilon^{i_1\cdots i_N}$ of SO(N) group can not appear in the four point function $\langle\phi_{i_1}(x_1)\phi_{i_2}(x_2)\phi_{i_3}(x_3)\phi_{i_4}(x_4)\rangle$ as long as $N\neq4$. (The $\epsilon^{i_1\cdots i_N}$ tensor appears in the N-point function.) After rescaling the $S$-channel OPE, the first three lines of the above crossing equation becomes exactly the $N=4$ case of (5). As we show in the main text, the constraints form the first three lines of crossing equation already allows detect the two dimensional $SU(2)_k$ WZW model.

As a final remark, the four point function of $SU(2)_1$ WZW model is,

$$\begin{aligned}\langle\phi_i(x_1)\phi_j(x_2)\phi_k(x_3)\phi_l(x_4)\rangle&=\frac{1}{|x_{12}||x_{34}|}\\&\quad\times\bigg(P_{ijkl}^{(0,0)}\frac{4(z-2)(\bar{z}-2)}{\sqrt{(z-1)(\bar{z}-1)}}\\&\quad+P_{ijkl}^{(1,1)}\frac{4z\bar{z}}{\sqrt{(z-1)(\bar{z}-1)}}\\&\quad+P_{ijkl}^{(1,0)}\frac{-4z\bar{z}+5\bar{z}+3z}{\sqrt{(z-1)(\bar{z}-1)}}+P_{ijkl}^{(0,1)}\frac{-4z\bar{z}+3\bar{z}+5z}{\sqrt{(z-1)(\bar{z}-1)}}\bigg)\,.\end{aligned}\quad(26)$$

In literature [49] it is often written in terms four point function of $SU(2)$ group elements,

$$\langle g(x_1)_{a_1}{}^{b_1}g(x_2)^{-1}{}_{b_2}{}^{a_2}g(x_3)_{a_3}^{-1}{}^{b_3}g(x_4)_{b_4}{}^{a_4}\rangle=\frac{1}{|x_{12}||x_{34}|}G(u,v)\,,$$

$$G(u,v)=z\bar{z}\sqrt{(z-1)(\bar{z}-1)}\bigg(\frac{\delta_{a_3}^{a_4}\delta_{a_1}^{a_2}}{z}+\frac{\delta_{a_3}^{a_2}\delta_{a_1}^{a_4}}{1-z}\bigg)\bigg(\frac{\delta_{b_4}^{b_3}\delta_{b_2}^{b_1}}{\bar{z}}+\frac{\delta_{b_4}^{b_1}\delta_{b_2}^{b_3}}{1-\bar{z}}\bigg)\,,$$

with $g$ being a $SU(2)$ group element and $a,b=1,2$.

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
