# Peer review of "Non-Wilson-Fisher kinks of $O(N)$ numerical bootstrap: from the deconfined phase transition to a putative new family of CFTs"

_SciPost Physics, doi:SciPost Phys. 10, 115 (2021)_

## Round 2 · Referee Report · Anonymous (Referee 1) · 2020-11-7

Strengths

1- The paper studies kinks in the bootstrap curve which do not correspond to Wilson-Fisher theories and in some cases they are able to identify them with a known theory. The goal of the conformal bootstrap is to chart out the space of CFTs using abstract principles, so this is an important step forward in this direction.

2- They further show, by working in non-integer dimension, that the kink disappears around d=2.7. This is consistent with previous work which says that the phase transition between a Neel magnetic ordered state and a valence ordered state is not described by a unitary CFT.

3- They give an analytic interpretation for the N goes to infinity limit of the new kinks and discuss the challenges in identifying it with a unitary CFT at large but finite N.

Weaknesses

1- There are places where the grammar and spelling can be improved.

2- The paper assumes background with the numerical bootstrap technology, which many readers may be unfamiliar with. Not all of the conventions are spelled out clearly.

3- There are some strange artifacts in the plot which do not appear to be physical, e.g. in the second plot of figure 3.

Report

This paper studies an interesting problem in the conformal bootstrap and meets the criteria to be published, with some small modifications.

Requested changes

1- The paper should mention the subtleties about working in non-integer d. It is well-known that theories, like the Wilson-Fisher model, are non-unitary in fractional dimensions. In principle the usual bootstrap machinery, which assumes unitarity, does not work. On the other hand, this non-unitarity appears to only affect high dimension operators and likely does not affect the main conclusions of this work. Nevertheless, this issue should be brought up.

2- There are a few places where grammar and spelling can be fixed. For example, on page 2 it should be "Interesting CFTs usually sit at“kinks” of the bootstrap curve ...". There are other minor typos, on page 3 "WZW term" and on pg 14 "As a final remark" are misspelled. These are minor issues, but should be fixed.

3- In figure 3 it would be useful if the k=1 theory was shown on the plot. That way the reader can see how close the theory is to saturation.

4- Finally, it may be useful to either give the conventions for Lambda and the normalization of the d-dimensional blocks explicitly or to reference an earlier paper that uses the same conventions.

---

## Round 2 · Referee Report · Connor Behan (Referee 2) · 2020-11-9

Strengths

  1. Gives newcomers a feel for how various bounds might look.
  2. Explains some features analytically.
  3. Discusses important future directions.
  4. Keeping the review minimal and jumping into results early is a good thing.

Weaknesses

  1. There are some typos that are easily fixed.
  2. Some familiarity with many previously obtained bounds is assumed.

Report

This paper advances our understanding of the bounds that the numerical bootstrap places on a four-point function of $O(N)$ fundamentals. Previous studies focused on external dimensions close to the unitarity bound where one can see "kinks" that describe well known Wilson-Fisher fixed points. In contrast, this paper studies "non-WF kinks" which were discovered more recently and appear at somewhat larger scaling dimension.

One main result is an exact solution for the $O(\infty)$ kink in $d$ dimensions. This is interesting as a starting point for understanding the non-WF kinks at finite $N$. It is also potentially useful in other bootstrap problems which do not appear to have considered this coset-like difference of correlators before. Another result shows that the $O(4)$ kink in 2 dimensions is described by a WZW model. The authors use this to study the Neel-VBS phase transition and find evidence that it is weakly first order, thus shedding light on an important question in condensed matter physics.

Overall the paper is a nice example of what can be learned by tuning various parameters in the conformal bootstrap. It should be published after a few corrections and clarifications are added.

Requested changes

  1. Correct "Jin-Beom Bae" in [24], "Di Francesco" in [25] and "Neel" in [33].
  2. Change "ect" to "etc" in the first paragraph of section 1.
  3. Change "a scalar operators" to "a scalar operator" in the fourth paragraph of section 1.
  4. Change "that we have good understanding" to "where we have good understanding" at the end of section 2.
  5. Change "almost saturate" to "almost saturates" in footnote 7.
  6. The sentence above equation 4.8 should read "This poses a puzzle that the theory seemingly contradicts a theorem [46] saying that CFTs with conserved higher spin currents ar e free theories which have a central charge proportional to N".
  7. Change "symmetry rank-2" to "symmetric rank-2" in the last paragraph of section 5.
  8. Change "relevant" to "irrelevant" in the sectond last paragraph of section 5.
  9. Change $SU(N)$ to $SU(N^*)$ and "whille" to "while" in the same paragraph.
  10. Change "its the action" to "its action" above A.2.
  11. A.7 should not have a period.
  12. Change "rank-2" to "the rank-2" in all figures and "below" to "bottom" in figure 4.
  13. Change "space-time dimensions $d$" to "space-time dimension $d$" everywhere.

  14. For the subtraction in equation 4.6, it sounds like the coefficients have been tuned so that some conformal blocks cancel between the two correlators. But do any OPE coefficients become negative? It would be good to say whether positivity of the expansion is proven to hold, believed to hold or known to break down at some order that's high enough to be negligible.

  15. It would be good to say that $N = N^{*2} - 1$ since the coincidence between $SU(N^*)$ and $O(N)$ bounds does not only hold asymptotically. Also, readers who are surprised by this might appreciate a note about whether this observation is purely empirical or follows from some known property of the crossing equations.
  16. Also in this part, you discuss the possibility of the non-WF kink being equivalent to one of the two kinks in the adjoint bootstrap. But it is not clear whether this is the leftmost kink (hypothesized to be QED3) or the rightmost kink (hypothesized to be QED3-GN).

---

## Round 3 · Referee Report · Connor Behan (Referee 2) · 2021-5-8

Report

The paper has undergone nice improvements and meets all the criteria to be published.

---

## Round 3 · Referee Report · Anonymous (Referee 1) · 2021-5-11

Report

The authors have made the suggested changes and I am happy to recommend the paper be published in its current form.

---

## Round 3 · Author Response

We would like to thank both referees for their comments on our manuscript. We have made the following changes based on the referee report.

---

## Round 3 · List of Changes

-We have corrected the grammar mistakes and typos pointed out by both referees. - In response to Requested Change 14 in Report 2, we added footnote 8 on page 10. - In response to Requested Change 15 in Report 2, we added footnote 10 on page 12. - In response to Requested Change 16 in Report 2: It is hard to say which of the two kinks does the non-WF kink corresponds to. In fact, it would be important to study the two kinks more carefully, especially to find a distinguishing operator content feature to confirm that they indeed correspond to two separate solutions of the crossing equation. We mentioned this in footnote 11. - In response to Requested Change 1 in Report 1, we added footnote 5 on page 7. - In response to Requested Change 3 in Report 1, we modified Figure 3. - In response to Requested Change 4 in Report 1: In the second to the last paragraph of section 1, we added a sentence to refer the definition of Lambda to an earlier paper. Since we did not present any explicit OPE coefficients in this paper, we figure that the convention of the block is not important therefore we will not mention it here.

---

## Editorial Decision

published